# The Evolution of Dietary Consumption in the Spanish Adult Population and Its Relationship with Environmental Sustainability

**DOI:** 10.3390/nu16244391

**Published:** 2024-12-20

**Authors:** Laura Álvarez-Álvarez, María Rubín-García, Facundo Vitelli-Storelli, Lorena Botella-Juan, Tania Fernández-Villa, Vicente Martín-Sánchez

**Affiliations:** 1Group of Investigation in Interactions Gene-Environment and Health (GIIGAS), Institute of Biomedicine (IBIOMED), University of León, 24071 León, Spain; lalva@unileon.es (L.Á.-Á.); mrubig@unileon.es (M.R.-G.); vitelli@uji.es (F.V.-S.); lbotj@unileon.es (L.B.-J.); tferv@unileon.es (T.F.-V.); 2Consortium for Biomedical Research in Epidemiology & Public Health, CIBERESP, Carlos III Institute of Health, 28029 Madrid, Spain; 3Perinatal Epidemiology, Environmental Health and Clinical Research, School of Medicine, Universitat Jaume I, 12071 Castellon, Spain

**Keywords:** dietary patterns, environmental footprint, greenhouse gases, household budget survey

## Abstract

Background/Objective: The relationship between food consumption and environmental sustainability is becoming increasingly evident. The aim of this study was to estimate the evolution of the environmental impact of food consumption in the Spanish population, assessed in terms of greenhouse gas (GHG) emissions. Methods: Data collected from the Household Budget Survey were included, from approximately 24,000 households for the period of 2006–2023. The environmental impact of diet, in terms of GHG emissions, was estimated from the EAT-Lancet Commission tables, and the adherence to the Mediterranean Diet (MedDiet) was calculated using the Dietary Score index. Results: The environmental impact of the Spanish diet, in terms of GHG, followed a downward trend over the years analysed, from 3978.1 g CO_2_-eq in 2006 to 3281.4 g CO_2_-eq in 2023, a decrease of 17.5%. The food groups with the largest decrease in consumption during this period were red meat (from 39.9 kg/year to 35.5 kg/year), fish (from 24.3 kg/year to 19.0 kg/year), and dairy products (from 113.4 kg/year to 99.7 kg/year). The level of adherence to the MedDiet increased slightly from 34 points in 2006 to 35 points in 2023 due to an increase in the amount of vegetables (42.7 kg/year vs. 44.3 kg/year) and grains consumed (53.1 kg/year vs. 72.6 kg/year) and a decrease in fish consumption (24.3 kg/year vs. 19.0 kg/year). Conclusions: In Spain, a reduction in GHG emissions associated with food consumption was observed due to a decrease in the consumption of red meat, fish, dairy products, and fats. National surveys are very useful tools to analyse the impact of food consumption on climate change and to assess the effect of the policies implemented to contain it.

## 1. Introduction

Climate change is considered one of the greatest public health threats of our time and, as such, is on the political agenda of international organizations, which stress the urgent need for global action to try to curb and reduce this problem [1]. The negative effects are not only limited to extreme weather events and the loss of biodiversity or natural resources [2] but also negatively affect health through food insecurity [3], an increase in pathogenic diseases [4], and even impacts on mental health [5].

One of the most relevant symptoms and causes of climate change is global warming, and, in turn, one of the main drivers of global warming is the human-induced increase in greenhouse gases in the atmosphere. Regardless of claims that climate change is the side effects of global warming or that global warming is a symptom of human-caused climate change, the discussion at hand is essentially about the same basic phenomenon: the build-up of excessive thermal energy in the Earth’s system [6,7]. In addition, food systems, according to the Intergovernmental Panel on Climate Change, are responsible for between 21% and 37% of GHG emissions [8]. This relationship raises the need to establish dietary patterns that improve the health of the environment while ensuring that we can feed the estimated 9 billion people expected to be alive in 2050 [9].

In the search for dietary patterns that are both healthy and environmentally friendly, governments in different countries have established expert committees to advise society on how to achieve these changes [10,11,12]. Among the diets recommended by these bodies, the Mediterranean Diet (MedDiet) stands out. This dietary pattern is considered sustainable by promoting the consumption of plant foods and reducing the consumption of animal products [13,14,15]. It is also recognized for its benefits in cardiovascular health [16,17] and the prevention of chronic diseases [18,19]. But, despite all the reported benefits of this dietary pattern, its adherence is declining in Mediterranean countries [20].

These data can be obtained from the information on food consumption that countries compile through different surveys, which are collected in large databases such as the European one, compiled by the European Food Safety Authority (EFSA) [21]. Within these surveys, in Spain, we focused on the Household Budget Survey (HBS), which is carried out annually by the National Statistics Institute and allows for the study of the consumption expenditure of families in this country [22].

Given that knowing and understanding the environmental impact of food consumption on the population is essential for the development of public policies to mitigate climate change, the aim of this study was to estimate the environmental impact, in terms of GHG emissions, of food consumption in the Spanish population between 2006 and 2023.

## 2. Materials and Methods

### 2.1. Data

Data from the HBS [22], collected over the period of 2006–2023, were used to carry out this study. The HBS allows us to ascertain the consumption expenditure of households residing in Spain, as well as the distribution of said expenditure among the different consumption divisions.

The HBS, published annually and including nearly 24,000 dwellings in its sample, provides information that is essential for estimating for the National Accounts the household expenditure on consumption, and for updating the Consumer Price Index (CPI) weightings.

The consumption expenditure that is recorded in the HBS refers both to the monetary flow that the household pays for certain final consumption goods and services, and to the value of certain non-monetary household consumption. Among the latter, it is appropriate to indicate the rent of owned dwellings (that is, the estimated rent of the dwelling owned by or granted to the household and in which the household resides), the salary in kind, the free or subsidised food or restaurant checks at the workplace, or the consumption originating from the production for their own final consumption (garden, farm, factory, workshop, or those taken from one’s own shop).

#### Sample

This survey collects, on an annual basis, socio-economic information on the standard of living and consumption of some 24,000 randomly selected Spanish households that collaborate for two consecutive weeks in each of the two years in which they form part of the sample. These households undertake to collect information on their expenditure for two weeks by means of questionnaires and interviews in which this information is supplemented.

The data collection is carried out in two ways:-An expenditure diary in which participating households record for 14 consecutive days all expenses incurred, including food purchased, detailing quantity, type and place of purchase;-Direct interviews to collect information on less frequent expenditures and general characteristics.

The classification used to analyse these data from 2006 to 2015 was COICOP (Classification of Individual Consumption according to Purpose) and, from 2016 onwards, ECOICOP (European Classification of Individual Consumption by Purpose), which was introduced to allow comparisons with other statistics such as the Consumer Price Index. These two classifications are structured in 12 major expenditure groups, and, in both cases, we used the first two categories, which refer to expenditure on food and beverage consumption, selecting the necessary items to be able to calculate the level of adherence to the MedDiet and the environmental impact of the diet in these households. These categories refer to food and beverage consumption expenditure in kg per person per year.

In the HBS, food is grouped into eight main food categories and two beverage categories, representing the main following consumption groups:-Bread and cereals: Includes bread, pasta, rice, flour, and other cereal products.-Meat: Divided into fresh meat (beef, pork, poultry, etc.) and processed meat (sausages and canned food).-Fish and seafood: Covers fresh, frozen, processed fish, and seafood products such as crustaceans and molluscs.-Milk, cheese and eggs: Includes liquid milk, dairy products (yoghurts, butter, and cream), and eggs.-Oils and fats: Includes vegetable oils (e.g., olive oil and sunflower oil) and animal fats.-Fruits: Classifies fresh, dried, and canned fruits.-Vegetables and tubers: Includes fresh, frozen, canned vegetables, and potatoes.-Other foods: Residual category for products such as sauces, spices, dietetic products, prepared meals, etc.-Non-alcoholic beverages: Includes products such as bottled water, soft drinks, juices, and other non-alcoholic beverages.-Alcoholic beverages: Includes wine, beer, and other alcoholic beverages.

The EPF is carried out in strict compliance with the General Data Protection Regulation (GDPR) of the European Union and the Organic Law on the Protection of Personal Data and Guarantee of Digital Rights (LOPDGDD) in Spain [23]. For more information on the HBS, the full methodology is available at https://www.ine.es/en/metodologia/t25/t2530p458_en.pdf (accessed on 15 July 2024).

### 2.2. Estimating Environmental Footprint

The EAT-Lancet Commission published a report in 2019, in which they developed global scientific targets for healthy diets from sustainable food systems. This report includes tables from a meta-analysis published by Clark et al. [24], in which they assessed the environmental impact of 81 foods according to five environmental impact factors, including GHG emissions (environmental footprint values of each food available in Appendix A). The methodology used to estimate these environmental factors was based on life cycle assessment (LCA) [25], which is standardised according to the International Organisation for Standardisation (ISO) and standardises environmental coefficients, allowing comparability between studies.

Using information on the environmental impact of food from the EAT-Lancet Commission tables [24,26] and food consumption data from the HBS, we estimated the impact of our sample’s diet in terms of GHG emissions as described elsewhere [27,28]. These calculations were carried out as follows:(1)Within each of the HBS food group (kg per person/year), the corresponding foods listed in the EAT-Lancet tables were included and converted to grams per person per day.(2)For items that did not refer to a single food (e.g., citrus fruits), the proportion consumed was calculated from available national survey data [29].(3)The GHG emissions of each food were obtained from the meta-analysis on which the EAT-Lancet recommendations were based [24], and the environmental impact was calculated by multiplying the value of the emissions of each food by the daily consumption of each product.(4)Finally, the environmental impact of the diet of the studied population, in terms of GHG emissions, was calculated as the sum of the contributions of each of the foods, considering the information collected in the HBS. GHG emissions were thus obtained in grams of CO_2_ equivalents per person per day (g CO_2_-eq/person/day).

In addition, the following data were taken into account for these calculations: the fish group included monkfish, turbot, sea bass, hake, sole, mackerel, salmon, trout, and tuna. The shellfish and molluscs group included mussels, squid, and prawns. Processed meats and deli meats were considered to be derived from beef and pork at 50% each.

### 2.3. Adherence to MedDiet

In order to calculate adherence to the MedDiet, we used the Dietary Score (DS) index proposed by Panagiotakos [30]. It includes 11 food groups (vegetables, potatoes, legumes, fruits, whole grains, fish, red meat, poultry, whole dairy, fats, and alcohol) and ranges from 0 to 55 points. The first six groups and the use of olive oil as a cooking fat score favourably, i.e., the higher the consumption, the higher the score, while meat, poultry, dairy, and alcohol score inversely, i.e., the lower the consumption, the higher the score. In addition, this index ranks adherence by tertiles, with the first tertile indicating low adherence and the third tertile indicating high adherence. The complete scoring can be found in Table 1.

Based on the information collected in the HBS, we grouped the items according to the DS group classification and calculated the servings/month or mL/day (in the case of alcohol) from the amount recorded in the survey, which was collected in kg/year.

### 2.4. Statistical Analysis

Both the foods from the EPF and the EAT-Lancet tables were grouped into 13 categories, resulting in the following food groups: fruits, vegetables, potatoes, legumes, grains, fish, meat, poultry, dairy, fats, sugar, and alcohol.

Based on the above classification, data were obtained for the Spanish population for the 17 years studied (2006–2023), both for the food groups consumed in each year (gr/person/day), adherence to the MedDiet (score 0–55), GHG emissions per food group, and the sum of total emissions (g CO_2_-eq/person/day).

A descriptive analysis of the variations in food consumption, adherence to the Mediterranean Diet, and GHG emissions was conducted throughout the study period.

GHG emissions averages were plotted to visualise changes over the years and to identify food groups with significant changes in consumption trends over the years analysed; Joinpoint Regression models were estimated by calculating annual percentage changes (APC). *p*-Values of less than 0.05 were considered statistically significant.

The R software version 4.1.1 [31] was used to determine the environmental impact of the diet, trends in food consumption were assessed using the Joinpoint Regression Program, version 5.3.0 [32], and the rest of the calculations were performed using Microsoft Excel.

## 3. Results

The information on the evolution of consumption in Spain between 2006 and 2023 can be found in Table 2. Among the changes observed between 2006 and 2023, there were decreases in the consumption of red meat (39.9 to 35.5 kg/year), fish (24.3 to 19.0 kg/year), dairy products (113.4 to 99.7 kg/year), fats (16.8 to 13.3 kg/year), and potatoes (27.6 to 22.6 kg/year), while there was an increase in the consumption of poultry (11.7 to 15.2 kg/year), eggs (117.8 to 150 units/year), and whole grains (53.1 to 72.6 kg/year).

These changes in food consumption had practically no effect on the MedDiet adherence, which remains almost unchanged, rising from 34 to 35 points. This shift in the score can be attributed to an increase in vegetable and cereal consumption, coupled with a reduction in fish intake, as evidenced in Table 2.

The changes in the consumption of the different food groups over this period were reflected in the reduction in annual GHG emissions from 3978.1 g CO_2_-eq in 2006 to 3281.4 g CO_2_-eq in 2023, which supposes a reduction of 17.5%. This decrease is mainly due to a decrease in the consumption of red meat, fish, seafood, and fats from 3415.4 to 2649.3 g CO_2_-eq/day (−22.4% in total). In contrast, the consumption of eggs, poultry, and dairy products increased over the years from 247.7 to 320.2 g CO_2_-eq/day (29.3% in total), while the consumption of other foods remained constant, varying only from 315.1 to 311.9 g CO_2_-eq/day (−1.0%) (Figure 1 and Figure 2, and Table 3).

Joinpoint models were applied to analyse time trends in the consumption of different food groups and to estimate APCs. As shown in Figure 3A, total GHG emissions decreased in two periods (APC_2006–2014_ = −1.42% and APC_2014–2017_ = −4.40%; *p* < 0.05 in both), then increased slightly (APC_2017–2023_ = 0.61%). The same trend is observed for GHG emissions from the group, consisting of red meat, fish and seafood, dairy, and fats (Figure 3B), decreasing significantly in two periods (APC_2006–2014_ = −1.81% and APC_2014–2017_ = −5.12%; *p*< 0.05 in both) and increasing slightly in the last period (APC_2017–2023_ = 0.60%). For the poultry and eggs group (Figure 3C), an increase was first observed (APC_2006–2012_ = 2.28%, *p* < 0.05), followed by a decrease (APC_2012–2019_ = −0.75%, *p* < 0.05) and a subsequent increase (APC_2019–2023_ = 3.01%, *p* < 0.05). Finally, the other food group (Figure 3D) showed an increase in the first period (APC_2006–2008_ = 2.24%) and a subsequent decrease (APC_2008–2023_ = −0.32%).

## 4. Discussion

This study analysed dietary sustainability in the Spanish population through changes in consumption patterns, using data from the HBS with records from 2006 to 2023. Our results showed a lower consumption of red meat, fish, dairy products, and fats, an increase in the consumption of chicken and eggs, and a constant consumption of plant-based products. In addition to the reduction in these products, this work has shown a trend towards a diet with a lower environmental impact in terms of GHG emissions.

Although we have not found any other article that jointly studies the evolution of consumption patterns in other countries and their relationship with environmental sustainability, different studies analysed the change in the consumption of certain food groups [33,34,35]. A study published by Kearney JM [34] analysed the change in the consumption of different food groups between 1980 and 2005, before our study, which started in 2006. It was observed that the consumption of all animal products increased, which is partially in line with the results of our study up to 2008, but from that year onwards in our study, it only increased for chicken and eggs. On the other hand, our results are partially consistent with those reported by Micha R et al. [33], who analysed dietary intake in 113 countries between 1990 and 2010. In this case, red meat consumption in Western European countries decreased in this period, which corresponds to the data obtained in our study. In contrast, their results showed that fish consumption in the same area increased in the years studied, while the opposite was found in our study.

The review by Dokova KG et al. [35], about the evolution from 1990 to 2020, reported similarities of the Spanish data of our study in terms of a decrease in carbohydrates (in our study, potato consumption decreases, but not grains), a decrease in saturated fats (in our study, red meat, an important source of saturated fats, decreases), and a decrease in fish consumption from 2010 to 2019.

Regarding environmental sustainability, according to our results, the reduction in the consumption of red meat, fish, seafood, and dairy was directly related to a reduction in GHG emissions, which is in line with several previously published articles stating that the reduction in the consumption of these food groups is associated with an improvement in this area [27,36,37,38,39,40]. These studies showed that a 50% reduction in meat substitution with legumes improved carbon footprint by 20% [41] and that meat substitution reduced GHG emissions by 34% [37]. The main reason for these findings is that livestock farming increases GHG, mainly carbon dioxide, methane, and nitrous oxide, due to the large amount of manure produced [42], in addition to using a large amount of agricultural land, which is directly related to increased soil degradation and biodiversity loss [43].

This is where plant-based diets become important in reducing the environmental impact of food, the more plant-based the diet is, the less it will impact the environment [44]. However, a Spanish study found that most of the sample followed a Mediterranean diet, with only 7.4% of respondents claiming to follow a plant-based diet (vegetarian, vegan, flexitarian) [45]. In this sense, although plant-based diets partly meet the criteria for sustainable diets [46] by reducing environmental impact, promoting health, and being economically viable, they do not meet it in the sense of being culturally acceptable. The Mediterranean diet meets all the criteria and is considered a World Heritage Site (UNESCO).

Although it is not possible to say for certain, there could be several reasons why these animal products continue to decline, mainly environmental concerns, health issues, animal welfare, and economic factors. Firstly, there are economic reasons since, as we have seen above, the amount of animal-based foods consumed in diets increases directly with income [47], and, in our study, the decline was observed to begin in 2008 and can, therefore, be related to the global economic crisis [48]. Secondly, increased awareness of animal suffering and environmental reasons also seems to affect the shift towards diets with fewer animal products [49].

Consumption habits evolve, and, even though different studies carried out in the Mediterranean area show how, in the last two decades, consumption patterns in these countries are changing towards less healthy diets [50,51,52], the results obtained in our study show that adherence to the MedDiet remains constant over the 18-year period, and even improves by 1 point from 2006 to 2023. The studies that report that adherence to the MedDiet has worsened the use of the Mediterranean diet pyramid as a reference [53] and compare it with what is being consumed in the Spanish population (2014–2015) [50]; alternatively, they justify it on the grounds that they consume less of the food groups that are characteristic of a traditional MedDiet in the 2000–2012 [54] and 2013 [55] study periods. We highlighted that maintaining good adherence to the Mediterranean diet is important, as adherence to this dietary pattern has been linked to a clear improvement in environmental sustainability [56,57,58].

### Strengths and Limitations

This study is not without limitations, some of which are inherent to how the data were collected. Although the HBS offers a broad view of food consumption in Spanish households [22], it does not take into account consumption outside the home, such as in bars or restaurants, so the intake of certain foods may be underestimated.

It should also be borne in mind that, during the period of analysis, events such as the 2008 economic crisis [59] or the COVID-19 pandemic [60] may have influenced consumption patterns temporarily, making it difficult to identify long-term trends.

Another important limitation is that there was no differentiation between age groups and socio-economic levels, both of which are important determinants of food choices [61,62].

Finally, as there is no standard method for calculating the environmental impact of food, the data we have obtained may not be comparable with other estimations. In addition, there may be products that are produced locally and, therefore, have a different environmental impact than the one used in our calculations, so we must be cautious as these are estimates.

However, the present study has several strengths. One is that, to our knowledge, it is the first time that household consumption data have been linked to environmental impact analyses, addressing not only changes in dietary habits over the years but also their implications for sustainability. Moreover, it is a work with a long analysis period over 18 years, which allows for identifying trends and changes in food consumption patterns.

Another strong point is the environmental impact database used, which works with the LCA technique and considers all the processes that take place from the time the food is produced until it reaches the consumer. Moreover, this method is standardised according to the International Organisation for Standardisation (ISO) and standardises environmental coefficients, allowing comparability between studies.

## 5. Conclusions

The results of this study showed how the diet of the Spanish population evolved over the years. This evolution resulted in a lower consumption of red meat, fish, seafood, and dairy products, which translates into an improvement in environmental sustainability in terms of reduced GHG emissions.

Specific studies are needed to better understand the population’s diet and better assess its impact on the environment as understanding changes in the population’s consumption habits is fundamental for the design of food and public health policies aimed at a more sustainable and healthy future.

Finally, we believe that this study may be useful in encouraging other countries to use their own national surveys to analyse not only food trends but also GHG emissions, using a methodology like the one we proposed. We believe that this is a useful tool for evaluating policies to reduce GHG emissions.

## Figures and Tables

**Figure 1 nutrients-16-04391-f001:**
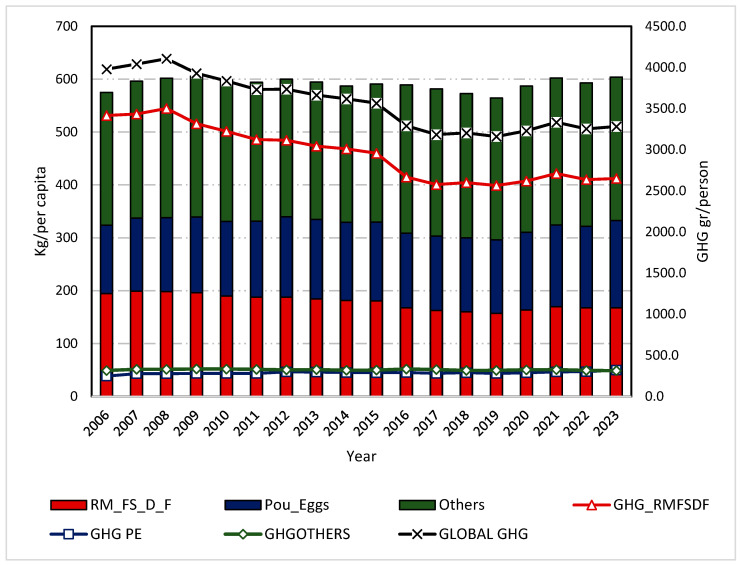
Evolution of the carbon footprint of Spanish households per food group and consumption by group. Note: GHG indicates greenhouse gas emissions; RM_FS_D_F, group of red meat, fish and seafood, dairy, and fats; Pou_Eggs, group of poultry and eggs; Others, group consisting of vegetables, fruits, legumes, potatoes, grains, sugar, and alcohol.

**Figure 2 nutrients-16-04391-f002:**
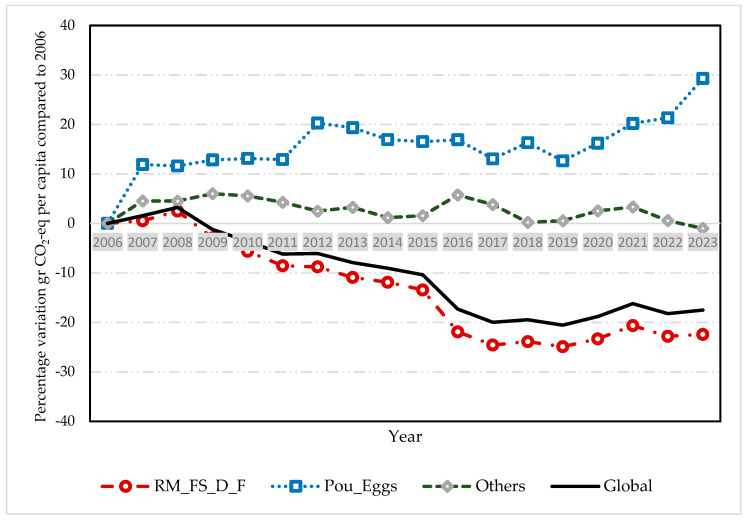
Change in the share of GHG emissions in Spanish households since 2006. Note: GHG indicates greenhouse gas emissions; RM_FS_D_F, group of red meat, fish and seafood, dairy, and fats; Pou_Eggs, group of poultry and eggs; Others, group consisting of vegetables, fruits, legumes, potatoes, grains, sugar, and alcohol.

**Figure 3 nutrients-16-04391-f003:**
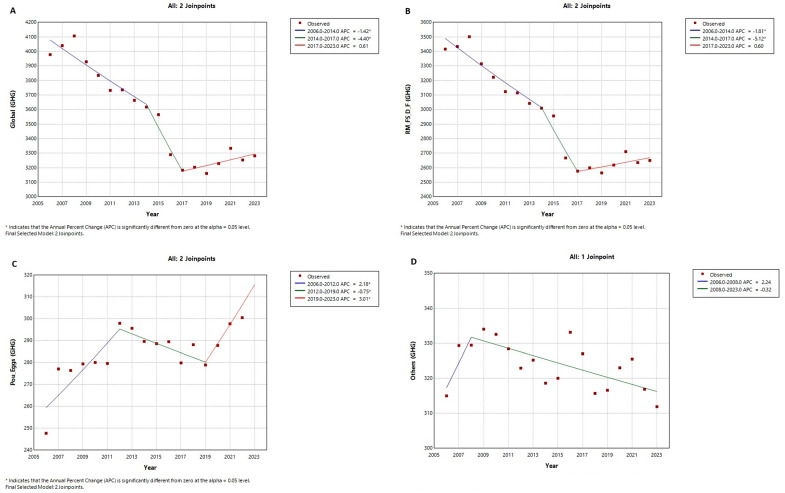
Joinpoint regression models for trends in GHG emissions from consumption of different food groups over the period 2006–2023. (**A**) Model for global GHG emission trends; (**B**) Model for trends in GHG emissions from red meat, seafood, dairy and fat consumption; (**C**) Model for GHG emission trends from poultry and egg consumption; (**D**) Model for GHG emission trends from consumption of vegetables, fruits, pulses, potatoes, cereals, sugar and alcohol.

**Table 1 nutrients-16-04391-t001:** Characteristics of the dietary score index to assess the adherences to MedDiet.

Food Groups	Dietary Score (0–55 Points)		
	Scoring		
Vegetables	All	0 -> 0 points 1–4 -> 1 point 5–8 -> 2 points 9–12 -> 3 points 13–18 -> 4 points >18 -> 5 points	Rations/month
Potatoes	All		
Legumes	All		
Fruit	All		
Grains	Whole grain		
Fish	All		
Meat	Red meat	0 -> 5 points 1–4 -> 4 point 5–8 -> 3 points 9–12 -> 2 points 13–18 -> 1 points >18 -> 0 points	
Poultry	All		
Dairy	With fats		
Fats	Olive oil	Never: 0p; hardly ever: 1p; ≤1: 2p; 1–3: 3p; 3–5: 4p; daily: 5p.	Use
Alcohol	All types	<300: 5p; 300–399: 4p; 400–499: 3p; 500–599: 2p; 600–699: 1p; >700: 0p.	mL/day

**Table 2 nutrients-16-04391-t002:** Annual evolution of consumption by Spanish households according to the different food groups and adherence to the MedDiet.

Kg/Year	2006	2007	2008	2009	2010	2011	2012	2013	2014	2015	2016	2017	2018	2019	2020	2021	2022	2023
Dairy	113.4	116.7	114.8	114.6	111.1	110.7	110.6	109.3	106.9	106.9	98.7	97.0	93.4	92.3	97.6	101.6	100.8	99.7
Fruits	76.9	81.6	77.9	77.9	81.2	79.1	77.9	76.7	75.8	80.2	79.6	80.3	80.9	78.1	76.8	81.3	74.9	75.2
Grains	53.1	56.0	55.1	55.5	54.3	54.0	52.6	52.3	51.8	50.6	75.5	73.7	72.4	72.4	70.0	71.8	73.6	72.6
Vegetables	42.7	44.7	45.0	45.4	46.3	47.2	46.0	47.7	46.7	47.3	46.6	45.6	44.3	43.3	47.0	46.8	44.8	44.3
Red Meat	39.9	40.8	43.2	40.8	39.5	38.8	39.5	38.5	38.5	38.2	34.8	33.3	34.3	33.9	33.9	34.6	34.2	35.5
Potatoes	27.6	21.2	29.9	31.1	30.1	27.3	28.4	28.8	28.8	27.2	24.1	24.4	22.2	21.7	22.7	21.7	21.3	22.6
Fish	24.3	24.7	23.9	23.8	23.1	22.7	22.4	22.1	21.3	21.3	20.5	19.3	19.1	18.7	19.5	20.5	19.6	19
Legumes	13.0	13.8	13.7	13.7	14.0	13.6	13.6	13.1	13.1	13.7	11.5	11.5	11.0	11.2	12.3	12.5	12.3	12.8
Alcohol	30.7	33.5	34.0	34.7	33.7	34.1	34.2	33.8	34.2	34.9	36.8	36.9	35.9	35.5	41.2	37.7	38.7	37.2
Poultry	11.7	13.3	13.2	13.3	13.4	13.3	14.2	14.1	13.8	13.7	14.0	13.4	13.5	13.4	13.7	14.1	14.3	15.2
Fats	10.8	7.8	11.3	11.2	10.6	10.6	10.5	9.5	9.6	9.1	8.7	8.3	8.9	8.3	8.2	8.7	8.9	7.8
Eggs (U)	117.8	125.0	126.8	129.7	127.9	130.1	137.5	136.2	133.9	135.1	127.2	127.4	126.5	125.4	133.5	140.4	139.5	150.0
Sugar	4.6	5.0	4.9	5.2	4.8	4.7	4.8	4.7	4.4	4.2	3.8	3.7	3.5	3.3	3.5	3.2	3.0	3.3
Dietary Score	34	35	35	35	35	35	35	35	34	34	35	35	35	35	35	35	35	35

Note: Egg consumption was registered in units/year.

**Table 3 nutrients-16-04391-t003:** Annual evolution of the carbon footprint (GHG/day) emitted by Spanish households according to the different food groups.

GHG/Day	2006	2007	2008	2009	2010	2011	2012	2013	2014	2015	2016	2017	2018	2019	2020	2021	2022	2023
Red Meat	1989.9	1979.4	2077.3	1893.9	1850.5	1766.7	1775.3	1740.8	1737.0	1694.5	1479.1	1436.6	1470.8	1450.0	1456.6	1487.6	1433.0	1465.2
Fish	585.3	581.0	549.0	541.0	525.6	508.1	498.5	484.3	465.7	461.9	448.2	410.2	404.4	403.8	426.2	450.6	414.5	401.2
Dairy	545.1	564.3	563.9	571.5	554.8	558.8	555.3	553.5	540.3	543.8	495.8	496.3	478.9	480.9	505.8	530.3	541.8	552.9
Fats	295.1	309.0	310.5	308.9	291.9	289.6	285.8	263.9	266.2	256.0	243.7	232.8	245.4	229.9	229.8	241.5	246.2	230
Poultry	189.1	214.9	213.3	214.9	216.5	214.9	229.5	227.9	223.0	221.4	226.2	216.5	218.2	216.5	221.4	227.9	231.1	245.6
Grains	119.7	126.2	125.0	126.3	122.6	122.0	119.3	118.9	117.4	114.9	131.4	127.5	125.5	126.6	122.9	125.6	129.1	127.5
Vegetables	74.0	76.8	77.2	77.7	79.3	79.9	77.4	80.6	78.1	78.0	77.1	74.5	72.3	70.4	75.7	75.2	72.1	69.8
Fruits	48.2	50.9	49.5	49.6	52.7	50.8	50.4	49.6	49.0	51.8	52.1	52.8	53.4	51.9	51.2	55.2	49.7	50.1
Alcohol	43.4	47.0	45.9	47.5	46.1	45.6	44.9	45.6	44.3	45.3	47.0	46.7	40.8	44.5	48.5	45.7	42.9	39.9
Potatoes	11.6	8.9	12.6	13.1	12.6	11.5	11.9	12.1	12.1	12.4	10.1	10.3	9.3	9.1	9.5	9.1	9.0	9.5
Legumes	7.8	8.3	8.2	8.2	8.4	8.1	8.2	7.9	7.8	8.2	6.9	6.9	6.6	6.7	7.4	7.5	7.4	7.7
Eggs	58.6	62.2	63.1	64.5	63.6	64.7	68.4	67.7	66.6	67.2	63.3	63.4	69.9	62.4	66.4	69.8	69.4	74.6
Sugar	10.3	11.2	11.0	11.6	10.8	10.5	10.8	10.5	9.9	9.4	8.5	8.3	7.8	7.4	7.8	7.2	6.7	7.4
Global	3978.1	4040.1	4106.4	3928.7	3835.4	3731.4	3735.7	3663.3	3617.4	3564.7	3289.4	3182.8	3203.2	3160.2	3229.2	3333.1	3252.8	3281.4

## Data Availability

Data described in the manuscript, including the code book and analytic code, are not publicly available due to privacy restrictions but can be available upon reasonable request to the corresponding author.

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
