# Peer review of "The Evolution of Dietary Consumption in the Spanish Adult Population and Its Relationship with Environmental Sustainability"

_nutrients, 2024, doi:10.3390/nu16244391_

Round 1
Reviewer 1 Report
Comments and Suggestions for Authors
Dear Authors
Below, I am providing comments and suggestions to enhance the quality of the manuscript:
INTRODUCTION
- The section on “Greenhouse gas (GHG) emissions” could be expanded further in the introduction to provide a clearer understanding of the issue.
METHODS
- This section is adequately presented; however, I believe it would be beneficial to clearly define the study design adopted in the first section. Additionally, ethical considerations and adherence to clinical best practices should be clearly outlined.
RESULTS
- I do not recommend changes to this section, as it is clear and well-structured. However, Figures 1 and 2 require a legend to clarify the acronyms used.
DISCUSSION
- In my opinion, the discussion could be expanded by including recent studies related to the Spanish population that evaluate dietary habits, such as “Plant-Based Diets versus the Mediterranean Diet” and their impact.
CONCLUSIONS
- The conclusions section is relatively brief. Could it be expanded? It would be helpful to include future implications of the study.
Additional Comments
- Please ensure consistency in the bibliography, using italics and bold where appropriate.
Author Response
-
INTRODUCTION
The section on “Greenhouse gas (GHG) emissions” could be expanded further in the introduction to provide a clearer understanding of the issue.
Following your suggestions, this section has been expanded to include some more information on this topic (page 2, lines 41-46).
- METHODS
- This section is adequately presented; however, I believe it would be beneficial to clearly define the study design adopted in the first section. Additionally, ethical considerations and adherence to clinical best practices should be clearly outlined.
Following their recommendations, we have expanded the section and included the ethical considerations of the survey (page 4, lines 127-129).
RESULTS
- I do not recommend changes to this section, as it is clear and well-structured. However, Figures 1 and 2 require a legend to clarify the acronyms used.
Thank you for your comments. Indeed, the legends for these figures were out of order and did not appear at the bottom of the figure. This has now been corrected in the manuscript.
DISCUSSION
- In my opinion, the discussion could be expanded by including recent studies related to the Spanish population that evaluate dietary habits, such as “Plant-Based Diets versus the Mediterranean Diet” and their impact.
One of the main problems we have had in carrying out this work is that we have not found studies in the Spanish population that analyse dietary habits and environmental impact with which to compare, but following your recommendations, we have expanded this section with regard to the theoretical aspects of plant-based diets (p. 10, lines 295-302).
CONCLUSIONS
- The conclusions section is relatively brief. Could it be expanded? It would be helpful to include future implications of the study.
We have completed this section by highlighting the future implications of the study (p.11,12, lines 355-356 and 359-363).
Additional Comments
- Please ensure consistency in the bibliography, using italics and bold where appropriate.
The bibliography has been corrected and unified in terms of format.
Reviewer 2 Report
Comments and Suggestions for Authors
I suggest the authors some revisions to improve the submitted manuscript:
Before the study’s main goals, a background statement should be included in the abstract. Future perspectives and some directions for further studies are also missing in this section.
The Introduction is not adequate. The paragraphs have no connection to each other and one statement paragraphs should be avoided. Besides of this, the content is very scarce. You need to provide a strong background and literature review to support the realization of your research.
The estimating environmental footprint needs to be better explained in the Materials and Methods section.
The tables are not well explained in the Results section. Please, elaborate on this.
The discussion is adequate but the Conclusions are very poor. Why your study is impactful and relevant? What directions can be followed based on the obtained results? Please, elaborate on this.
Author Response
Comments and Suggestions for Authors
I suggest the authors some revisions to improve the submitted manuscript:
Before the study’s main goals, a background statement should be included in the abstract. Future perspectives and some directions for further studies are also missing in this section.
Following your recommendations, we have added a background statement and expanded the conclusion section with future perspectives (page. 1, lines 14-17).
The Introduction is not adequate. The paragraphs have no connection to each other and one statement paragraphs should be avoided. Besides of this, the content is very scarce. You need to provide a strong background and literature review to support the realization of your research.
Following your recommendations, we have restructured the introduction and added more bibliographical data (page. 2, lines 41-46).
The estimating environmental footprint needs to be better explained in the Materials and Methods section.
We have completed this section with additional information on the calculation of the environmental impact as suggested (page 4, lines 137 and 158-161).
The tables are not well explained in the Results section. Please, elaborate on this.
We have completed the results section with new analyses that we believe help in understanding the results (page 9, lines 245-256).
The discussion is adequate but the Conclusions are very poor. Why your study is impactful and relevant? What directions can be followed based on the obtained results? Please, elaborate on this.
Thank you for your valuable comment. We have expanded the conclusions, highlighting the importance of this study in terms of methodological replicability, allowing us to create a global perspective on this issue. In addition, we emphasise the importance of diet as a tool to fight climate change (p.11,12, lines 355-356 and 359-363).
Round 2
Reviewer 1 Report
Comments and Suggestions for Authors
The authors have provided substantial revisions to the manuscript. It is now ready for publication.
Reviewer 2 Report
Comments and Suggestions for Authors
The manuscript has been improved considerably.